



# Spatially varying parameters improve carbon cycle modeling in the Amazon rainforest with ORCHIDEE r8849

Lei Zhu[1,2,3,4], Philippe Ciais[3*], Yitong Yao[5*], Daniel Goll[3], Sebastiaan Luyssaert[6], Isabel Martínez Cano[3], Arthur Fendrich[3,7], Laurent Li[4], Hui Yang[8], Sassan Saatchi[9], Wei Li[1,2*]

[1]Department of Earth System Science, Ministry of Education Key Laboratory for Earth System Modeling, Institute for Global Change Studies, Tsinghua University, Beijing, China
[2]Ministry of Education Ecological Field Station for East Asian Migratory Birds, Beijing, China
[3]Laboratoire des Sciences du Climat et de l'Environnement, LSCE/IPSL, CEA-CNRS-UVSQ, Université Paris-Saclay, Gif-sur-Yvette, France
[4]Laboratoire de Météorologie Dynamique / IPSL, CNRS, Sorbonne Université, Ecole Normale Supérieure - PSL Université, Ecole Polytechnique - Institut Polytechnique de Paris, Paris, France
[5]Division of Geological and Planetary Sciences, California Institute of Technology, Pasadena, CA, USA
[6]Faculty of Science, A-LIFE, Vrije Universiteit Amsterdam, Amsterdam, the Netherlands
[7]European Commission, Joint Research Centre (JRC), Ispra, Italy
[8]College of Urban and Environmental Sciences, Key Laboratory for Earth Surface Processes of the Ministry of Education, Peking University, Beijing, China
[9]Jet Propulsion Laboratory, California Institute of Technology, Pasadena, CA, USA

*Correspondence to*: Philippe Ciais (philippe.ciais@lsce.ipsl.fr), Yitong Yao (yyao2@caltech.edu) and Wei Li (wli2019@tsinghua.edu.cn)

**Abstract.** Uncertainty in the dynamics of Amazon rainforest poses a critical challenge for accurately modeling the global carbon cycle. Current dynamic global vegetation models (DGVMs), which use one or two plant functional types for tropical rainforests, fail to capture observed biomass and mortality gradients in this region, raising concerns about their ability to predict forest responses to global change drivers. Here we assess the importance of spatially varying parameters to resolve ecosystems spatial heterogeneity in the ORCHIDEE (ORganizing Carbon and Hydrology in Dynamic EcosystEms) DGVM. Using satellite observations of gross primary productivity (GPP), tree aboveground biomass (AGB) and biomass mortality rates, we optimized two key parameters: the alpha self-thinning ($\alpha$), which controls tree mortality induced by light competition, and the nitrogen use efficiency of photosynthesis ($\eta$), which regulates GPP. The model incorporating spatially optimized $\alpha$ and $\eta$ parameters successfully reproduces the spatial variability of AGB ($R^2$=0.82), GPP ($R^2$=0.79), and biomass mortality rates ($R^2$=0.73) when compared to remote sensing observations in intact Amazon rainforests, whereas the model using spatially constant parameters has $R^2$ values lower than 0.04 for all observations. Furthermore, the relationships between the optimized parameters and ecosystem traits, as well as climate variables were evaluated using random forest regression. We found that wood density emerges as the most important determinant of $\alpha$, which are in line with existing theory, while water deficit conditions significantly impact $\eta$. This study presents an efficient and accurate approach to enhancing the simulation of Amazonian carbon pools and fluxes in DGVMs by assimilating existing observational data, offering valuable insights for future model development and parameterization.



## 1 Introduction

More than half of the global rainforests are located in the Amazon basin, storing approximately 140 PgC in their living biomass (Pan et al., 2011; Vancutsem et al., 2021). They constitute an important carbon sink in intact forests, but deforestation and forest degradation can turn these areas into a carbon source (Brienen et al., 2015; Gatti et al., 2021). The Amazon rainforest is

characterized by high levels of biodiversity, rainfall variability, and soil diversity (Malhi et al., 2004; Ter Steege et al., 2006; Flores. et al., 2010; Quesada et al., 2010; Castanho et al., 2013). These factors are only partly resolved in dynamic global vegetation models (DGVMs), which are widely used to predict vegetation responses to global environmental changes (Prentice et al., 2007). Therefore, current DGVMs can hardly capture continental scale carbon fluxes and stock spatial gradients (Johnson et al., 2016), leading to even higher uncertainty in projecting future carbon dynamics in the Amazon rainforests.

The factors driving the spatial variability of carbon fluxes and stocks in the Amazon rainforests are complex and remain largely unresolved (Johnson et al., 2016; Muller-Landau et al., 2021). Field forest plots reveal a clear aboveground biomass (AGB) gradient: higher AGB in the northeast over the Guiana Shield, where nitrogen fixing, large-seeded legumes trees with lower growth rates are dominant, and lower AGB in the drier southwest, where non-legume trees thrive (Ter Steege et al., 2006; Malhi et al., 2006; Mitchard et al., 2014). The biomass gradient results from the balance between the spatial variation in woody

productivity and carbon losses, mainly through tree mortality. Field observations suggest a strong association between the spatial variation of tree mortality and the gradients of AGB (Malhi et al., 2015; Johnson et al., 2016). In western and southern regions, where natural disturbances are more frequent, dominant species tend to have higher growth rates, lower wood density, and higher mortality rates, supporting the "grow fast, die young" life history strategy, maintaining stable biomass (Baker et al., 2004; Malhi et al., 2006; Keeling and Phillips, 2007; Esquivel-Muelbert et al., 2020; Brienen et al., 2020). Woody net primary

productivity (NPP) is influenced by both climate conditions and soil fertility, particularly total soil phosphorus content which is higher in the western region. However, the mortality rate may be driven by drought, windblown disturbances, and soil physical properties including soil depth, soil structure, topography, and anoxic conditions (Malhi et al., 2004; Quesada et al., 2012; Sousa et al., 2022; Feng et al., 2023).

Current DGVMs cannot capture the spatial heterogeneity in productivity, biomass, and mortality rates in the Amazon. While

most models include detailed photosynthetic modules, they often ignore nutrient constraints on plant growth (e.g. phosphorus; Reed et al., 2015), and they simplify varied life history strategies to a few plant functional types with static parameters (e.g. mortality rates; Johnson et al., 2016). Although an increasing number of models have incorporated phosphorus cycles (Goll et al., 2012; Wang et al., 2010), simplified tree demography (Moorcroft et al., 2001; Rödig et al., 2017; Koven et al., 2020; Naudts et al., 2015), and drought induced mortality (Yao et al., 2022), their capability to capture these spatial heterogeneities

remain limited. More importantly, most models generally use one or two plant functional type (PFT) with uniform parameters across the entire Amazon forest. Building upon these setups, in DGVMs, variations in biomass carbon pools across different locations are primarily driven by differences in climate forcing and soil properties. However, biomass variability in the Amazon rainforests cannot be fully explained by climate factors alone, and soil properties are inadequately represented in DGVMs due





to the coarse resolution and simplified processes (Quesada et al., 2012; Galbraith et al., 2013; Saatchi et al., 2015; Joetzjer et

al., 2022). Given the availability of various data for parameter estimations, applying a single set of PFT-specific parameters uniformly across the entire basin is no longer justifiable (Butler et al., 2017).

Several studies have attempted to simulate the spatial variations of biomass and GPP in the Amazon forest by linking different processes or forcing the model with observed parameters. In models that apply a constant tree mortality rate, higher NPP is typically associated with higher biomass (Keeling and Phillips, 2007). Delbart et al. (2010) demonstrated that linking higher

tree mortality rates to higher NPP in a DGVM can improve the simulation of spatial biomass variations, but may fail to capture current biomass sinks in pristine tropical forests (Brienen et al., 2015). Castanho et al. (2013) forced another DGVM with an interpolated map of observed woody biomass residence times and correlated maximum carboxylation rate ($V_{cmax}$) with a soil total phosphorus map. While this study successfully reproduced site-level woody NPP and AGB, the use of interpolated woody biomass residence times and soil phosphorus data, based on limited observations, remains questionable (Saatchi et al., 2015;

He et al., 2023).

The increasing availability of remote sensing data provides wall-to-wall observations of key forest properties such as AGB, GPP, tree height, leaf area index (LAI), and normalized difference vegetation index (NDVI), which all remain underutilized for parameterizing and evaluating models. Rödig et al. (2017) developed an individual-based model, using a remote-sensing canopy height map as a constraint, and linked mortality rates to soil clay fraction and precipitation to simulate the spatial

gradient of biomass. Ma et al. (2024) incorporated more observations from satellite including GPP, LAI, AGB and forest age, and applied a stepwise calibration at the global scale. Despite these advancements, few studies have simultaneously employed multiple remote sensing datasets to optimize their models in the Amazon forest.

In this study, we aim to improve the modeled spatial gradient of biomass, productivity and mortality across the intact Amazon rainforest in the ORCHIDEE (ORganizing Carbon and Hydrology in Dynamic EcosystEms) model. This model simulates

forest demography based on tree cohorts and incorporates tree mortality mechanisms, including self-thinning and drought-induced mortality (Yao et al., 2022). We assessed whether spatial variation in model parameters can improve the ability of the ORCHIDEE model to simultaneously reproduce multiple observed patterns, including AGB, GPP, and biomass mortality rates derived from remote sensing data. Finally, we discuss mechanistic links between environmental factors and spatial gradients in model parameters by exploring the results of an explainable machine-learning method, in order to guide future model

developments.

## 2 Materials and methods

In this study, we describe the optimization of the model parameters for the Amazon forest (Plant functional type: tropical broadleaf evergreen tree) using remote sensing data. We first set a baseline model with optimal, but spatially constant parameters at the Amazon basin scale. Next, we allowed parameters to vary in space with a spatial resolution of 1°×1°. Finally,

we used a random forest model to derive relationships between the potential influencing factors and the optimal spatially



varying parameters. This approach enabled us to explain the emerging spatial variations of the parameters across the Amazon basin and prescribe their spatial variability in the model (Fig. 1).

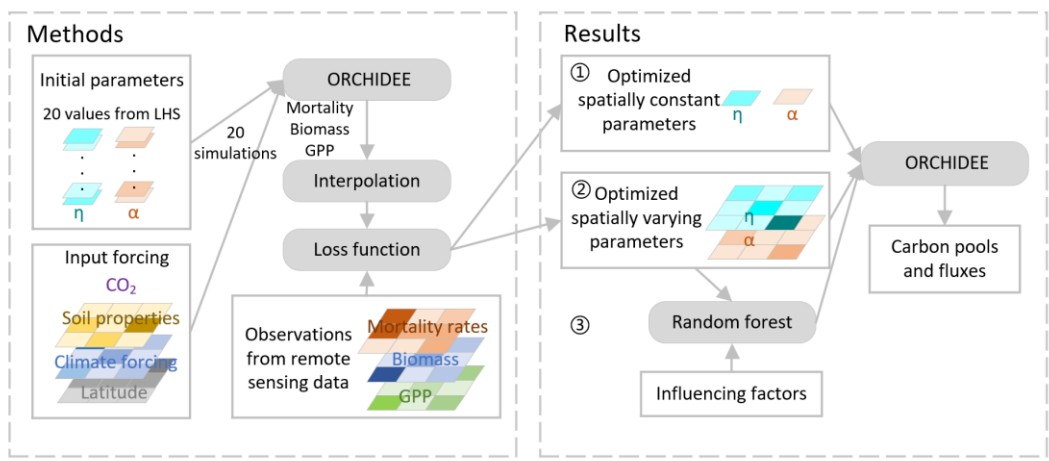

**Figure 1: Flowchart for the parameter optimization. η: nitrogen use efficiency; α: alpha self-thinning; LHS: Latin hypercube sampling; GPP: gross primary productivity.**

## 2.1 Study area

This study focuses on the Amazon basin (79°W-43°W, 18°S-10°N), predominantly covered by old-growth, intact tropical evergreen rainforest (Flores et al., 2024). We excluded regions based on the following criteria: 1. areas with mean annual precipitation below 1,500 mm and mean annual temperature below 18 °C based on CRUJRA v2.4 from 1901-1920 (Climatic Research Unit and Japanese Reanalysis data; Harris et al., 2020) to keep the focus on the warm and humid tropics; 2. grid cells (1°×1°) with less than 10% forest cover, based on forest pixels that are labeled as "undisturbed tropical moist forest" in TMF (Tropical Moist Forest; Vancutsem et al., 2021) to exclude that pixels that were likely to be regrowing or degraded.

## 2.2 The ORCHIDEE model

ORCHIDEE is a global terrestrial ecosystem model that simulates energy, water, and carbon cycles (Krinner et al., 2005; Naudts et al., 2015; Vuichard et al., 2019). In this study, we used ORCHIDEE r8240 and further incorporated soil-tree-leaves hydraulics and drought-induced tree mortality (Yao et al., 2022). We also adjusted the NPP allocation parameters and forced the model with a spatially explicit wood density map (See Supplementary text 1, Table S1; Chave et al., 2010; Yang et al., 2024; Chen et al., 2020; Yang et al., 2021). These developments collectively constitute the new ORCHIDEE version r8849.
The input data for the ORCHIDEE model include soil texture and meteorological forcing (air temperature, precipitation, shortwave downward radiation, longwave downward radiation, specific humidity, wind speed and atmospheric pressure). We used CRUJRA v2.4 with a 6-hour temporal resolution as the source of meteorological forcing (Harris et al., 2020). To reduce computational time, we resampled the original CRUJRA data from a resolution of 0.5°×0.5° to 1°×1°. Consequently, all results are presented at 1°×1°resolution. For soil texture data, we used the Harmonized World Soils Database version 2.0 (HWSD2;



Figure S1; FAO and IIASA, 2023), a comprehensive global soil database that includes various national datasets at a resolution

of 1 km. In the ORCHIDEE model, soil texture determines the fractions of clay, silt, and sand, with other soil properties derived from a lookup table (Table S2) that control soil hydraulic conductivity and soil water holding capacity. Soil depth is prescribed as 2 m in the model. Given that more than 99% forest in our study area is tropical broad-leaved evergreen trees (Harper et al., 2023), we forced the model with only tropical broad-leaved evergreen trees PFT in each pixel. Here, we provide a summary of the model processes. All changed parameters are summarized in Table S1.

**2.2.1 GPP**

In the ORCHIDEE model, GPP is calculated based on the leaf scale model of Farquhar et al. (1980) further simplified by Yin and Struik (2009). The key parameters driving the photosynthesis process are $V_{cmax}$ (µmol $CO_2$ $m^{-2}$ $s^{-1}$) and $J_{max}$ (the maximum electron transport rate, µmol $CO_2$ $m^{-2}$ $s^{-1}$), with $J_{max}$ typically assumed to be proportional to $V_{cmax}$. In ORCHIDEE model, $V_{cmax}$ is dynamically calculated as follows:

$$V_{cmax} = \frac{\eta \times M_{n,l}^{active}}{\sum_{i=1}^{n} ((1-0.7 \times (1-Light_i)) \times LAI_i)} \tag{1}$$

where $M_{n,l}^{active}$ (gN $m^{-2}$) is the active nitrogen content in the leaves, $Light_i$ is the fraction of cumulative light transmitted to canopy level i, and $LAI_i$ ($m^2$ $m^{-2}$) is the leaf area index at level i. The ORCHIDEE model utilizes a dynamic three-dimensional canopy structure model to simulate gap probability (Haverd et al., 2012). The crown is divided into n levels (n=10). At each level, the model calculates the leaf volume based on tree crown morphology and density. $LAI_i$ is then calculated by leaf mass,

specific leaf area (SLA=0.0153 $m^2$ $gC^{-1}$), and leaf volume for this layer. The canopy gap is calculated based on $LAI_i$, tree height, tree density, and solar altitude angle. $Light_i$ decreases progressively from 100% at the top level to the gap fraction at the bottom level. Given the limited observations available to constrain the nitrogen cycle in the Amazon forest and prevailing limitation by geogenic soil nutrients like P or K, we instead prescribe leaf nitrogen concentration for the whole amazon forest (Sitch et al., 2003). In addition, we deactivate nitrogen controls on ecosystem processes and associated soil-plant feedbacks

(Supplementary text 2).

**2.2.2 Biomass mortality rates**

We defined biomass mortality rates as the ratio of biomass loss due to self-thinning to the total biomass prior to mortality. The ORCHIDEE model incorporates 20 circumference tree size classes within each forest PFT (Naudts et al., 2015), thus facilitating the representation of annual recruitment of young individual trees, and tree mortality driven by light competition

(i.e. self-thinning). Self-thinning-induced tree mortality is calculated as Eq. (2):

$$N_{max} = \left(\frac{D_g}{\alpha}\right)^{\frac{1}{\beta}} \tag{2}$$

where $N_{max}$ is the maximum number of trees per hectare for a given quadratic mean tree diameter ($D_g$, cm), $\alpha$ (cm) and $\beta$ are the self-thinning parameters. The model calculates relative density index (RDI) as the ratio of the number of the trees to $N_{max}$.



When the RDI exceeds the upper threshold of RDI (RDI$_{upper}$), the excess trees are removed through mortality evenly in each

circumference class, until the RDI reaches the lower threshold of RDI (RDI$_{lower}$). Both RDI$_{upper}$ and RDI$_{lower}$ are cubic functions of D$_g$, with user-defined parameters. This self-thinning Eq. (2) indicates that the growth of some trees occurs at the expense of others, which die due to competition for limited resources such as water, light, or nutrients (Yu et al., 2024). The power-law function assumed in the model has been widely observed in the field plots (Enquist and Niklas, 2001; Muller-Landau et al., 2006). Higher values of α and β suggest greater environmental carrying capacity and lower biomass mortality rates. In the

model, the default value for β is -0.73, which is close to the average value reported by Yu et al. (2024) for tropical forests. However, the default value for α has not been evaluated and to our knowledge, no empirical estimates for α have been provided in previous studies. We thus estimate the α based on field forest inventory data and find the average value is 1941 cm (Brienen et al., 2015).

The model also incorporates a drought-induced tree mortality module (Yao et al., 2022) to resolve periodic increase in mortality

associated with droughts. When drought-induced mortality exceeds the background mortality from the self-thinning process, the model assigns the total mortality rate to the drought-induced value; otherwise, it defaults to the background mortality rate. The drought mortality parameters were calibrated previously for the drought experiments in Caxiuanã National Forest (Yao et al., 2022). Although drought-induced mortality can reduce biomass periodically in addition to self-thinning, AGB and biomass mortality rates are predominantly controlled by α rather than drought-induced mortality at the basin scale during our

study period (Supplementary text 1.3).

### 2.2.3 Aboveground biomass

ORCHIDEE model has nine biomass pools: aboveground sapwood (Sap$_{ab}$), belowground sapwood, aboveground heartwood (Heart$_{ab}$), belowground heartwood, leaves, fruits, labile carbohydrate (Carbo$_{lab}$), and reserved carbohydrate (Carbo$_{res}$). The modeled aboveground biomass is calculated as follows:

$$AGB = Sap_{ab} + Heart_{ab} + Leaf + Fruit + \frac{Carbo_{lab} + Carbo_{res}}{2} \qquad (3)$$

### 2.3 Observational data

We used GPP observations from GOSIF for data assimilation at the grid scale (Li and Xiao, 2019). GOSIF products estimated GPP using sun-induced chlorophyll fluorescence (SIF) from GOSAT, GOME-2, and OCO-2 at a 0.05° resolution from 2001 to 2022. Given the high uncertainty in GPP estimates, particularly in tropical forests where GPP observations are scarce, we

also evaluated other GPP products derived from flux tower upscaling (FLUXCOM_RS and FLUXCOM_RS_METEO) and process-based models (BESSv2 and FORMIND) as part of our sensitivity tests (Jung et al., 2020; Li et al., 2023; Rödig et al., 2019). Across these datasets, the average GPP values for the Amazon basin range from 24.0 to 33.4 MgC/ha/yr. Although GOSIF provides the highest estimation, it appears to be more consistent with observed GPP values (Figure S3, Marthews et



al., 2012; Malhi et al., 2015). We compared GPP products with model outputs from 2001 to 2015 for GOSIF, FLUXCOM_RS,
FLUXCOM_RS_METEO, and BESSv2. For FORMIND, the GPP represents average values from 2003 to 2006.

We use the biomass map from Yu et al. (2023) to benchmark AGB in the ORCHIDEE model in 2020 (hereafter, Yu-Biomass). The Yu-Biomass map is based on a large number of lidar-biomass models, field plot data, and L-band radar data, with an original resolution of 100 m. Several other biomass maps are available for this region (Santoro and Cartus, 2023; Avitabile et al., 2016; Baccini et al., 2012). These maps generally incorporate field observation data and multiple remote sensing datasets,
including radar backscatter, LiDAR, and optical indices. Among these maps, Yu-Biomass map shows the highest correlation with independent airborne LiDAR estimates in the Amazon forest (Longo et al., 2016; Yu et al., 2023). We also validated these datasets using field observations (Mitchard et al., 2014) and found that the Yu-Biomass map captures the biomass gradient more accurately than the others (Fig. S4).

We optimized the modeled biomass mortality rates using biomass mortality data derived from Planet data (Dalagnol et al.,
2023; Dalagnol et al., 2021). This dataset estimates forest gaps based on very high-resolution (about 4-5 m) remote sensing and airborne lidar scanning data processed through deep learning methods. The identified gaps are then converted to biomass mortality estimates. The dataset provides annual biomass mortality data from 2016 to 2019 aggregated at a 1 km resolution. Given that this study focuses on the background mortality, to minimize the potential influence of the 2015/16 El Niño drought, we excluded the year 2016 and used the average from 2017 to 2019. Biomass mortality rates were derived by dividing biomass
mortality by total aboveground biomass from Yu-Biomass. We also compared the remote sensing-based biomass mortality rates to stem mortality rates from 189 long-term RAINFOR forest plots, with data primarily collected between 1981 and 2010 (Esquivel-Muelbert et al., 2020). This comparison yielded an R² of 0.39 and an RMSE of 0.53% (Fig. S5).

## 2.4 Simulation protocol

We began with a spin-up simulation to bring the vegetation biomass carbon pool to equilibrium, i.e., linear trend in vegetation
carbon over 40 years of < 0.03 MgC/ha/yr for 99% of pixels. An equilibration of the soil biogeochemistry was not needed because of absence of any effect on vegetation processes. We recycled the climate forcing from CRUJRA during 1901-1920 with a constant $CO_2$ concentration of 296.8 ppm (i.e. the $CO_2$ concentration in 1901). Starting from this equilibrium state, we proceeded with the transient simulation using climate forcing and $CO_2$ concentration data for the period 1901-2020. We did not account for land cover change, as our analysis is limited to undisturbed forests.

## 2.5 Optimization processes

Based on the description of the ORCHIDEE processes in 2.2, we conducted a sensitivity analysis of various model parameters on AGB, GPP, and biomass mortality rates using one-at-a-time approach. This method is widely used for sensitivity analysis, where each parameter is varied individually while keeping the others unchanged (Liu et al., 2015). We define the $VP_i$ (variation percentage, %) as an indicator for the sensitivity of model output variables i (i.e. AGB, GPP, and biomass mortality rates).
Following a previous study (Liu et al., 2015), we varied each parameter by ±10%, and VP is calculated as follows:



$$VP = \frac{|run_{+10\%} - run_{-10\%}|}{run_{ref}} \times 100 \qquad (4)$$

where $run_{+10\%}$ and $run_{-10\%}$ are the model results with +10% and −10% variations in a given parameter while fixing other parameters, $run_{ref}$ is the reference model result with default parameters. We selected parameters related to self-thinning, photosynthesis, carbon allocation and turnover rate in the model. AGB and mortality rate is the most sensitive to β, α, and η, while GPP is most sensitive to η, SLA, and $k_{lsmin}$ (Table 1).

**Table 1 Variation percentage (VP) of AGB, GPP and mortality rate for different parameters**

| Parameters | Description | $VP_{AGB}$ | $VP_{GPP}$ | $VP_{Mor}$ |
|---|---|---|---|---|
| α | Coefficient of the self-thinning relationship | 26.70% | 1.08% | 45.53% |
| β | Coefficient of the self-thinning relationship | 124.29% | 9.52% | 248.05% |
| η | Nitrogen use efficiency of Vcmax | 16.81% | 34.22% | 36.20% |
| SLA | Specific leaf area | 7.64% | 23.28% | 16.13% |
| $k_{rcon}$ | Hydraulic conductivity of roots | 3.25% | 3.71% | 6.77% |
| $k_{scon}$ | Hydraulic conductivity of sapwood | 3.23% | 3.69% | 6.77% |
| $k_{lsmin}$ | minimum observed leaf area to sapwood area ratio | 5.75% | 16.64% | 9.87% |
| $k_{lsmax}$ | maximum observed leaf area to sapwood area ratio | 0.88% | 2.59% | 1.54% |
| $k_{\tau s}$ | longevity of sapwood | 0.74% | 8.48% | 0.97% |
| $k_{\tau l}$ | longevity of leaf | 3.07% | 3.59% | 6.39% |
| $k_{\tau r}$ | longevity of root | 0.39% | 0.45% | 0.84% |

To optimize the model parameters for matching the spatial pattern of GPP, AGB, and biomass mortality rates, we constructed a loss function to optimize two key parameters: α and η. We chose the two parameters for the following reasons: 1) although β is more sensitive to AGB and mortality rate than α, α has a wider range of values compared to parameter β (Brienen et al., 2015; Yu et al., 2024). Regardless of which parameter is optimized, the observations can be matched. To avoid the issue of equifinality when optimizing both parameters, we chose to adjust α while keeping β fixed at its default value, considering that α is less constrained than β in previous studies (Yu et al., 2024). 2) η has the highest sensitivity to GPP in the model and is found to be highly variable within the moist tropical biome (Kattge et al., 2009; Ellsworth et al., 2022). 3) we do not try to optimize more parameters because the number of optimized parameters should not exceed the number of observations (GPP, AGB and biomass mortality rates) to avoid the equifinality issue.





We used the Latin hypercube sampling (LHS) method to select the initial parameter values, a technique widely used in model parameterization (Mckay et al., 2000; Yan et al., 2023). LHS can subdivide full parameter space evenly to generate representative parameter values. This procedure ensures an efficient sampling of the parameter distribution with a suitable
amount of parameter samples, thus avoiding impracticable computational efforts due to numerous model simulations (Mckay et al., 2000). The recommended number of sampled points is typically 10 times the number of dimensions (Loeppky et al., 2009). Accordingly, we used 20 samples as the initial parameter values, generated using LHS. The default η value is 14.08 μmolCO$_2$/gN/s based on the biome average of trait data for tropical forests on highly weathered soils (Kattge et al., 2009) and the α value is 1941 cm (Brienen et al., 2015). In this study, the initial uniform prior distribution ranges for α and η were set to
1000-2800 cm and 8-18 μmolCO$_2$/gN/s, respectively, to ensure that all optimized values fell within these boundaries (Fig. S7). After the initial ORCHIDEE run based on the 20 sets of parameter values, we interpolated the results for GPP, AGB, and biomass mortality rates over the whole domain using quadratic splines (Fig. S8), considering that these variables change monotonously and continuously with α and η. The interpolation achieved R$^2$ values of 0.99, 0.98, and 0.93 for AGB, GPP, and biomass mortality rates, respectively, based on leave-one-out validation, where 19 simulations were used for interpolation and
the remaining one for validation.

Next, for each 1°×1° grid cell, we constructed a quadratic loss function to identify the best estimates of α and η that minimize the loss (Groenendijk et al., 2021), defined as:

$$Loss_{agb}(\alpha,\eta) = (\frac{AGB_{sim}(\alpha,\eta) - AGB_{obs}}{\overline{AGB_{obs}}})^2 \tag{5}$$

$$Loss_{gpp}(\alpha,\eta) = (\frac{GPP_{sim}(\alpha,\eta) - GPP_{obs}}{\overline{GPP_{obs}}})^2 \tag{6}$$


$$Loss_{mor}(\alpha,\eta) = (\frac{Mor_{sim}(\alpha,\eta) - Mor_{obs}}{\overline{Mor_{obs}}})^2 \tag{7}$$

$$Loss(\alpha,\eta) = \frac{Loss_{agb}(\alpha,\eta)}{CV_{AGB}} + \frac{Loss_{gpp}(\alpha,\eta)}{CV_{GPP}} + \frac{Loss_{mor}(\alpha,\eta)}{CV_{mor}} \tag{8}$$

where AGB$_{sim}$(α, η), GPP$_{sim}$(α, η), and Mor$_{sim}$(α, η) are the AGB, GPP, and biomass mortality rates simulated by ORCHIDEE model after interpolation; AGB$_{obs}$, GPP$_{obs}$, and Mor$_{obs}$ are the AGB, GPP, and biomass mortality rates from the observations; $\overline{AGB_{obs}}$, $\overline{GPP_{obs}}$ and $\overline{Mor_{obs}}$ are the spatial mean AGB, GPP, and biomass mortality rates for the whole basin; CV$_{AGB}$, CV$_{GPP}$,
and CV$_{Mor}$ are the coefficients of variation for AGB, GPP, and biomass mortality rates from the observations. We used coefficients of variation as weight to balance different Loss from AGB, GPP, and biomass mortality rates at regional scale. We selected the α and η values that minimized Loss(α, η) for each grid cell and generated spatial maps of both parameters across the entire Amazon region. Then, we re-ran the ORCHIDEE model forced by the α and η spatial maps obtained from the optimization. For comparison, we also performed the same optimization using spatially constant α and η values across the
entire Amazon region (Fig. 1).



## 2.6 Analyze the influencing variables

Previous empirical studies have found that GPP/NPP and mortality rates are influenced by various factors including climate conditions, wood density, water table depth, and soil properties such as soil texture and soil fertility (Malhi et al., 2004; Quesada et al., 2012; Vicca et al., 2012; Castanho et al., 2013; Campioli et al., 2015; Sousa et al., 2022). We thus selected eight variables to explain the spatial distribution of the optimal $\alpha$ and $\eta$ values (Fig. S9, Table S3). Mean annual temperature (MAT), mean annual precipitation (MAP), and downward shortwave radiation (SWdown) are obtained from CRUJRA v2.4 (Harris et al., 2020). Maximum cumulative water deficit (MCWD) is a proxy for climatic water supply and indicates the intensity of drought. MCWD is calculated as:

$$CWD_t = \begin{cases} P_t - E_t, & if\ P_t < 100\ mm\ and\ t = 1 \\ CWD_{t-1} + P_t - E_t, & if\ P_t < 100\ mm\ and\ t > 1 \\ 0, & if\ P_t \geq 100\ mm \end{cases} \tag{9}$$

$$MCWD = \min(CWD_t) \tag{10}$$

with t being the month 1, …, 12, $P_t$ is the precipitation for month t calculated from CRUJRA-v2.4, and $E_t$ is the monthly evapotranspiration, which is fixed at 100 mm (Aragão et al., 2007). MCWD was calculated based on the hydrological year, which begins in May (t=1) and ends in April (t=12) of the following year (Chen et al., 2024). Wood density was sourced from Yang et al. (2024). Clay fraction from HWSD2 serves as a proxy for soil texture (FAO and IIASA, 2023). Soil total phosphorus content, a key indicator of soil fertility, was derived from Darela-Filho et al. (2024). Water table depth, representing both anoxic conditions and additional water availability beyond precipitation, was obtained from Fan et al. (2013). To avoid multicollinearity, precipitation was excluded from the analysis due to its high correlation with MCWD (Fig. S10).

We used random forest regression to analyse the potential influences of the eight variables on $\alpha$ and $\eta$. Random forest regression is a widely used machine learning algorithm that builds multiple decision trees to improve prediction accuracy (Breiman, 2001). We randomly selected 75% of the data to train the model, leaving the remaining 25% for validation. The random forest model was optimized with the following parameters for both $\alpha$ and $\eta$: ntree=100, max_depth=3. We constructed 50 models using different random seeds and averaged their outputs to derive the final results.

SHAP (SHapley Additive exPlanations) values were used to evaluate the contribution of each explanatory variable in the random forest regression (Lundberg et al., 2020). Based on Shapley values in game theory, SHAP values represent the marginal contribution of each variable, considering all possible combinations of feature values (Shapley, 1953). A SHAP value, whether positive or negative, indicates a corresponding positive or negative effect on the model's output. This method offers an interpretable insight into how individual features impact model predictions. We built the random forest model using the Python package "sklearn" and computed SHAP values with the Python package "shap".





## 3. Results

### 3.1 Simulations with optimal spatially constant parameters

290

The optimal spatially constant parameter values for $\alpha$ and $\eta$ are 1898 cm and 12.14 $\mu molCO_2/gN/s$, respectively, to match the AGB, GPP, and biomass mortality rates from observations. Basin-scale averaged values of AGB, GPP, and biomass mortality rates closely aligned with observations ($126 \pm 7$ MgC/ha, $33 \pm 3$ MgC/ha/yr, and $2.2 \pm 0.2$ % vs. $127 \pm 29$ MgC/ha and $33 \pm 2$ MgC/ha/yr and $2.0 \pm 0.6$ %, respectively). However, simulations failed to capture spatial heterogeneity, such as the higher

295 biomass in the Guiana Shield or the higher biomass mortality rates in the southeastern Amazon (Fig. 2). The $R^2$ values between the simulated AGB, GPP, and observations data were less than 0.04, and the correlation between simulated biomass mortality rates and observations was even negative (Fig. 3a-3c). Climate forcing, wood density, and soil texture had limited impact on modeled spatial variation of AGB, resulting in flat gradients of forest AGB and biomass mortality rates. In the ORCHIDEE model, higher MCWD is generally associated with higher GPP (Fig. 3b), but in reality, GPP doesn't vary significantly across

300 the entire Amazon region (Fig. 2b). On the contrary, MCWD has little influence on modeled tree mortality, despite its observed association with higher mortality rates (Fig. 3c). These discrepancies suggest that spatially varying parameters are necessary to better match the modeled AGB, GPP, and mortality rates with observational data.







**Figure 2: Spatial patterns of AGB, GPP, and mortality rates from the model with spatially constant parameters (a-c), the model with spatially varying parameters (d-f), and the observations (g-i).**



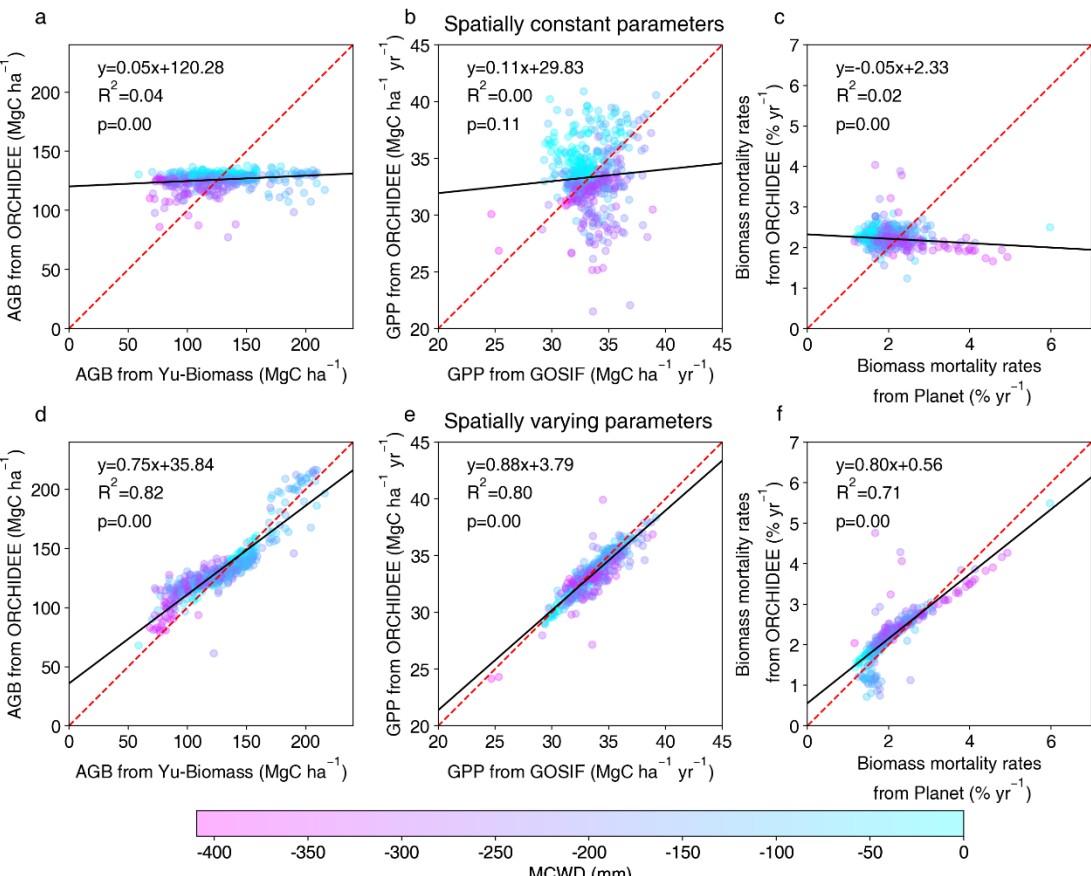

**Figure 3: Comparison of model-simulated AGB, GPP, and mortality rates with observations. Results from model simulation with spatially constant parameters (a-c) and model with spatially varying parameters (d-f). The dashed red line is the 1:1 line. The black solid line is the best fit between modeled results and observations. The color of the dots represents the value of MCWD to show its correlation with AGB, GPP, and biomass mortality rates.**

### 3.2 Simulation with optimal spatially varying parameters

The model with optimal spatially varying parameters successfully captured the spatial patterns of AGB, GPP, and mortality rates, including the higher biomass in the Guiana Shield, the lower GPP in the northwestern Amazon, and the higher mortality in the southern Amazon (Fig. 2). The R² values between the observed data and model-simulated AGB, GPP, and mortality rates improved substantially, increasing from near zero to 0.82, 0.79, and 0.73, respectively (Fig. 3). The spatially varying α values were higher in the northeast and lower in the southeast, while η values were higher in both the northeast and southern regions (Fig. S11).

Optimizing both α and η is necessary. When only α is optimized, the model-simulated AGB and mortality rates improved a lot compared with observations, but GPP shows a weak correlation with observed GPP (R²=0.01, Fig. S12a-c). This is mainly because GPP is not sensitive to α (Table 1). Similarly, VP of η is larger for GPP and biomass mortality rates than AGB (Table 1). Optimizing only η results in no correlation between model-simulated AGB and observed AGB (R²=0.0, Fig. S12d-f),





although simulated GPP and mortality rates had a stronger correlation with the observations. The spatial pattern of α shows a strong positive correlation with Yu-Biomass and a negative correlation with observed mortality rates. In contrast, η is positively
correlated with GOSIF GPP and mortality rate data (Fig. S13).

### 3.3 Spatial Correlation between the optimal parameter maps and influencing factors

We further applied explainable machine learning (i.e., random forest regression and SHAP) to find the influence of various factors on the spatial pattern of optimal parameters. The seven variables explained 45% of the variation in α and 48% in η on testing data. The Shapley results show non-linear relationships between the optimal parameters and influencing factors,
including climatic, plant, and edaphic properties. For example, when MCWD is below -300 mm, α increases sharply with rising MCWD, but once MCWD exceeds -300 mm, its impact on α becomes minimal. Similarly, the negative impact of MCWD on η becomes positive when MCWD falls below -100 mm (Fig. 4b). Clay fraction has a greater impact on η when it exceeds 40%, particularly in soils classified as clay or silty clay, which account for 35% of all grids (Fig. 4e; Fig. S1; Table S2). Wood density shows a strong positive correlation with α when it exceeds 0.61 gC/cm$^3$ (Fig. 4d), while the effect of water table depth
on α follows a dome-shaped pattern, where both deep and shallow water table depths reduce α (Fig. 4g).

Wood density shows the highest feature importance for α, based on mean absolute SHAP values (Fig. 5a). Additionally, MCWD emerges as the most important feature for η (Fig. 5b). When we re-run the model using parameters predicted by the random forest regression, it still captures the spatial gradients of carbon pools and fluxes, though with reduced accuracy compared to the optimized results. The modeled AGB, GPP, and mortality rates explain 47%, 25%, and 22% of the spatial
variation in the observations, respectively (Fig. S14).



**Figure 4: SHAP values of α and η for 7 explanatory variables: (a) temperature; (b) maximum cumulative water deficit (MCWD); (c) downward shortwave radiation (SWdown); (d) wood density; (e) clay fraction; (f) total available phosphorus; (g) water table depth. The shaded area indicates the standard deviation of SHAP values. The grey bars represent the distributions of explanatory variables.**



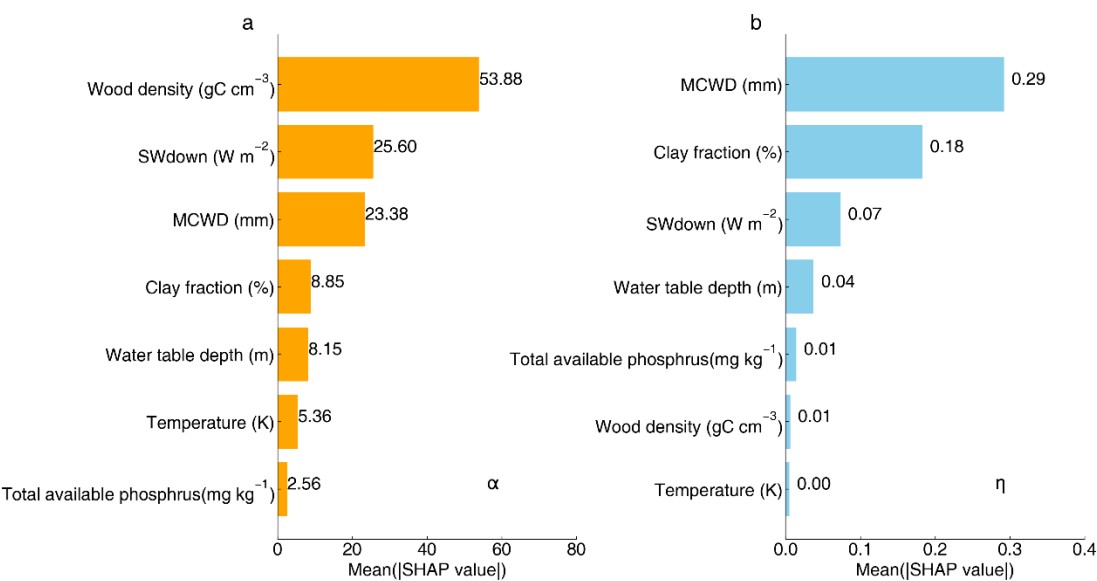

**Figure 5: Variable importance characterized by the mean SHAP values of 7 explanatory variables affecting α (a) and η (b).**

## 4. Discussion

**4.1 Implications from the inconsistency between model results and observations.**

Although our model shows much better agreement with the observations, some grid cells still exhibit underestimation or overestimation of AGB, GPP, or biomass mortality rates (Fig. S15), which reflects trade-offs made during the optimization process of the loss function. In the ORCHIDEE model, GPP, biomass, and mortality rates are tightly coupled. The amount of dead woody biomass is determined by both biomass and mortality rates, which should equal the fraction of NPP allocated to 355 wood under an equilibrium state. While carbon use efficiency (CUE=NPP/GPP: 33.7% ± 1.2%) and the woody NPP fraction ($f_{NPP}$: 30.8 ± 1.1%) don't vary a lot in the model, a grid with low biomass and low mortality rate should exhibit a correspondingly low GPP (Fig. S16). When observations are inconsistent with each other (e.g., low biomass, low biomass mortality rate but high GPP from observations), optimizing α and η alone is insufficient to accurately simulate AGB, GPP, and biomass mortality rates simultaneously.

There are two potential reasons for the remaining model-observation inconsistency. First, the model may overlook important processes. For instance, the model currently only accounts for woody biomass loss through tree mortality. However, Zuleta et al. (2023) found that 42% of woody biomass loss occurs due to damage to living trees, such as branch falls, trunk breakage, and wood decay. In the ORCHIDEE model, aboveground woody biomass is divided into two pools: sapwood and heartwood. To improve the accuracy of biomass mortality estimates, a separate biomass pool for branches should be incorporated in future 365 model developments to better account for biomass losses from branches. Besides, while CUE and $f_{NPP}$ fall within observed ranges, the variability of these factors may be underestimated. Some studies suggest that CUE ranges from 23% to 45%



(Doughty et al., 2018), whereas our model estimates it at 32% to 42%. Similarly, the spatial standard deviation of observed $f_{NPP}$ (2.9%) exceeds that of our modeled results (1.1%; Yang et al., 2021).

Second, the observational data may contain inherent biases. Mortality rates derived from field observations may be
overestimated because field plots are often located near roads and rivers, where disturbances are more frequent (Saatchi et al., 2015). Additionally, satellite AGB may be underestimated in dense forests due to sensor saturation (Yu et al., 2023). GPP data carries the greatest uncertainty, largely due to the limited observations in the Amazon basin. We compared five GPP datasets across the Amazon basin (see Methods). While GOSIF and FLUXCOM_RS datasets show higher GPP in the southwestern Amazon and lower GPP in the central region, the spatial patterns differ across other regions and datasets (Fig. S17). The
highest correlation coefficient between the GPP datasets is 0.46, observed between GOSIF and FLUXCOM_RS, while correlations among the other datasets are generally weaker (Fig. S3b).

We conducted sensitivity tests using additional GPP and biomass datasets. Four other GPP datasets were used to optimize the parameters α and η. After optimization, both AGB and GPP showed strong agreement between the model and observations, demonstrating the robustness of our approach (Fig. S18). The linear correlations between the eight influencing factors and the
optimal parameters are generally consistent with previous results (Fig.s S19, S20), with only a few exceptions. For example, when using GPP data from FLUXCOM_RS+METEO, MCWD and wood density are positively correlated with η, which is different when using GOSIF and other GPP datasets.

**4.2 Implications for ORCHIDEE model from the random forest regression**

The uncertainties in simulated carbon stocks and fluxes arise from model structure, parameterization, and forcing data (Bonan
and Doney, 2018). During the optimization of α and η, we assumed that all uncertainties stem from parameterization. By analyzing the correlation between the optimal parameters and influencing factors, our results showed great consistency with existing knowledge and highlighted important processes that are either missing or poorly represented in the ORCHIDEE model when simulating the Amazon forest.

For instance, a threshold of -300 mm in MCWD was identified in the partial dependence analysis between α and MCWD.
When MCWD drops below -300 mm, further decreases lead to lower α, indicating higher mortality (Fig. 4b). This result aligns with previous findings showing a negative correlation between AGB mortality rates and MCWD only when MCWD is particularly low (Sousa et al., 2022). Although our model incorporates drought-induced tree mortality, the higher tree mortality under lower MCWD conditions is likely underestimated. The dome-shaped relationship between water table depth and α is also in line with earlier studies (Fig. 4g). Shallow water table depths are associated with low biomass, as excessive water can
limit oxygen flow to roots, restricting plant growth (Costa et al., 2023). In waterlogged conditions, weakened root systems and reduced soil cohesion increase the risk of tree falls (Ferry et al., 2010). Conversely, deeper water table depths elevate the risk of drought-induced mortality, thereby increasing mortality rates (Chen et al., 2024). However, water table depth is not explicitly represented in our model. We also found that higher wood density is associated with higher α, which is in line with previous findings that forests with higher wood density tend to have lower mortality rates and higher biomass (Figure 4d;





Keeling and Phillips, 2007; Brienen et al., 2020). However, our model does not account for the feedback between wood density
       and tree mortality rates.

       Furthermore, we observed that η increases when MCWD falls below -100 mm. In the model, reduced water availability limits
       transpiration, leading to a decrease in GPP (Yao et al., 2022). However, the observed GPP exhibits a weak negative correlation
       with MCWD (Fig. S21). This suggests that grid cells with more negative MCWD require a higher η to simulate higher GPP.

Given the unclear mechanisms driving the correlation between MCWD and GPP in the Amazon forest, it remains uncertain
       whether η should increase as MCWD decreases. We also found that η increases when the clay fraction exceeds 40%. Although
       precipitation is distributed evenly across soil types with varying clay fractions, the runoff fraction increases sharply when the
       clay fraction surpasses 40% (Fig. S22), due to the very low hydraulic conductivity of clay soils (Table S2). Besides, while
       available phosphorus is thought to influence woody production (Quesada et al., 2012), total phosphorus does not seem to be a

critical factor in explaining the variation in optimal α and η (Fig. 5). In fact, the correlation between total available phosphorus
       and GPP is quite weak (R²=0.05), and the correlations with AGB or mortality rates are even weaker (Fig. S23). This may be
       due to uncertainties in the GPP data (Fig. S3, S17).

### 4.3 Strengths of the optimization method

       In this study, we significantly improved the simulations of spatial gradients and mean values of AGB, GPP, and biomass
mortality rates in the Amazon forest by optimizing two key parameters related to photosynthesis and self-thinning. Our method
       accounted for parameter variations within a PFT (tropical broad-leaved evergreen trees), an important factor that has largely
       been overlooked in other models (Butler et al., 2017). Besides, unlike previous studies that optimized numerous parameters
       simultaneously, our approach provided an efficient parameterization framework by focusing on the sensitive parameters for
       each pixel. In the future, we can incorporate additional observations, such as LAI and tree height, as their accuracy improves.
This method is also applicable to other DGVMs. Except for optimizing the parameters, this approach also helps identify the
       missing or under-represented processes in the model. We also tried to link the optimal parameters with other plant traits or
       environmental factors. By doing this, we may eventually improve the model by incorporating relationships between different
       traits (e.g., linking wood density to α).

### 4.4 Limitations for this work

Our parameter optimization framework lacks the ability to capture temporal variation accurately due to data insufficiencies.
       Besides, the input datasets used for model calibration carry significant uncertainties. More importantly, even with accurate
       observation data, our optimized parameter could be incorrect, as the mismatch between model results and observations can
       arise from model structure and forcing data (Bonan and Doney, 2018).

       In addition to the missing processes mentioned earlier, this version of ORCHIDEE model also lacks several key processes,
such as phosphorus carbon interactions and specie-based functional groups diversity (Reed et al., 2015; Sun et al., 2021; Rüger
       et al., 2020). We ignored the spatial variability of leaf nitrogen content by fixing it uniformly across the Amazon forest.



However, $V_{cmax}$ depends on both η and the leaf C/N ratio, the latter also exhibiting significant variability (Ellsworth et al., 2022).

In the context of climate change, the Amazon forest is increasingly exposed to disturbances such as droughts, fires, and
windthrow events (Chen et al., 2024; Flores et al., 2024; Feng et al., 2023). Although the model accounts for drought-induced mortality, it does not include other disturbances like fire, windthrow, and logging. In this study, we focused exclusively on intact forests, as defined by TMF data (Vancutsem et al., 2021). Incorporating natural and anthropogenic disturbances to this ORCHIDEE version is underway (Naudts et al., 2015; Yue et al., 2014; Marie et al., 2024; Chen et al., 2018) and might turn out to be essential in future work to fully capture the complexity of ecosystem dynamics across the entire Amazon rainforest.

**5. Conclusion**

Our study developed an efficient parameter optimization framework that integrates observation-based data for AGB, GPP, and mortality rates to optimize the spatial variation of parameters in the ORCHIDEE model. By incorporating spatially varying parameters, the model effectively simulates the spatial variation of AGB, GPP, and mortality rates. The spatial patterns of the optimal parameters are generally reasonable, revealing previously under-represented processes in the model. Given the
complexity of the Amazon forest and the practical limits on model complexity, applying spatially varying parameters can help for accurately representing its spatial variability.

**Acknowledgements**

This study was supported by the CALIPSO (Carbon Loss In Plants, Soils and Oceans) project, funded through the generosity of Eric and Wendy Schmidt by recommendation of the Schmidt Science program. Lei Zhu acknowledges support from the
China Scholarship Council Program (No. 202206210184).

**Code availability**

The ORCHIDEE (r8849) code used in this study is open source and distributed under the CeCILL (CEA CNRS INRIA Logiciel Libre) license. It is deposited at https://forge.ipsl.jussieu.fr/orchidee/wiki/GroupActivities/CodeAvalaibilityPublication/ORCHIDEE-Amazon with guidance
to install and run the model at https://forge.ipsl.jussieu.fr/orchidee/wiki/Documentation/UserGuide. The source data and code for Figure 2-5 are available via Zenodo at https://doi.org/10.5281/zenodo.15023110.

**Data availability**

Yu-Biomass map data (Yu et al., 2023) is available from https://doi.org/10.5281/zenodo.7583611. GOSIF data (Li and Xiao,
2019) can be downloaded from http://data.globalecology.unh.edu/data/GOSIF_v2/. Wood density map (Yang et al., 2024) can



be downloaded from Global patterns of tree wood density (zenodo.org). CRUJAR data (Harris et al., 2020) can be downloaded from https://catalogue.ceda.ac.uk/uuid/aed8e269513f446fb1b5d2512bb387ad/. HWSD2 data (FAO and IIASA, 2023) can be downloaded from https://data.isric.org/geonetwork/srv/api/records/54aebf11-ec73-4ff8-bf6c-ecff4b0725ea. Soil total phosphorus data (Darela-Filho et al., 2024) can be downloaded from https://doi.org/10.25824/redu/FROESE. Water table

depth data (Fan et al., 2013) can be downloaded from http://thredds-gfnl.usc.es/thredds/catalog/GLOBALWTDFTP/catalog.html.

**Author contribution**

PC and WL designed the research. LZ performed analysis and wrote the paper. YY merged plant hydraulic module to this version. All authors contributed to commenting on and writing the manuscript.

**Competing interests**

The authors declare no competing interests.

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
