# Peer review of "Spatially varying parameters improve carbon cycle modeling in the Amazon rainforest with ORCHIDEE r8849"

_EGUsphere, 2025_

## Author Comment (AC4)

**Response to comments**

**Paper #:** *EGUSPHERE-2025-397*

**Title:** *Spatially varying parameters improve carbon cycle modeling in the Amazon rainforest with ORCHIDEE r8849*

**Journal:** *Geoscientific Model Development*

**Reviewer #2:**

**Comments:**

**Comment #1**

The authors presented spatially explicit parameter tuning to improve a DGVM simulation of biomass, GPP and mortality gradient in intact Amazon forests. The work is very comprehensive and the results make sense for me. I have only minor comments below for the authors to consider.

**Response #1**

We thank the reviewer for the positive assessment of our manuscript. Please see our detailed point-by-point responses below.

**Comment #2**

The Introduction would benefit from an enhanced discussion on how the spatial gradient of morality rates was driven by different types of mortality, e.g., drought, wind and self-thinning as optimized in this study.

**Response #2**

We thank the reviewer for the suggestions. We revised the sentence on **L56-58** in the main text:

"The mortality rate could be driven by the intensity of forest disturbances, particularly windthrow and drought. In the western region, windstorms are the primary cause of tree death, while in the east-central and southern regions, drought is the dominant driver (Esquivel et al., 2020). "

**Comment #3**

Line 117: How wood density influences carbon cycle/mortality processes in ORCHIDEE? I noticed section 1.2 in Supplement but still it does not explain the physiological impacts of wood density.

**Response #3**

We revised the text in **Supplementary Text 1.2** to clarify how wood density influences the carbon cycle and mortality processes in the ORCHIDEE model:

"In the ORCHIDEE model, wood density is a prescribed trait/parameter that influences both carbon allocation and tree diameter. A higher wood density leads to a lower value of the stem-to-leaf allocation factor $f_{KF}$ (Eq. S3), which reduces leaf biomass and consequently lowers photosynthetic capacity. In addition, higher wood density results in smaller tree diameters and increased stem density (Eq. 2). Although empirical studies have shown that higher wood density is often associated with lower mortality rates (Esquivel-Muelbert et al., 2020), this relationship is not currently represented in the ORCHIDEE model."

References:

*Esquivel-Muelbert, A., Phillips, O. L., Brienen, R. J. W., et al. (2020). Tree mode of death and mortality risk factors across Amazon forests. Nature Communications, 11(1), 5515. doi:10.1038/s41467-020-18996-3*

**Comment #4**

Line 164-170: what is this 'background mortality'? I would think this is just self-thinning before it is represented in the model. Now that we already have self-thinning explicitly represented, why we still need this 'background mortality'? How its magnitude is determined?

**Response #4**

In our model, we use self-thinning mortality to represent background mortality, as self-thinning mortality is the dominant mortality process. This background mortality is mainly due to the resource competition among trees.

To clarify this point, we have added the following sentence on **L172–174**:

"Considering that self-thinning is the dominant and regularly occurring mortality process, we treat self-thinning mortality as background mortality driven by resource competition among trees in this study."

**Comment #5**

Line 204: the spinup process was already influenced by the parameters governing the self-thinning mortality and hence there is the issue of circularity here. How is this considered/addressed?

**Response #5**

We thank the reviewer for raising this point. We would like to clarify that the model parameters were not modified during the spin-up or transient simulations. The parameter optimization in this study was conducted to improve the simulation of GPP, AGB, and biomass mortality rates under present-day conditions. Since there are no observational constraints available for the end of the spin-up period, we did not attempt to calibrate the model for that phase. It is also a common practice in similar modeling studies to re-run the spin-up using optimized parameters (Peylin et al., 2016; Ma et al., 2024).

We added sentences on **L213-216** to clarify this point:

"Model parameters remain unchanged during the spin-up and transient simulations. Each time parameters are modified, we re-run the spin-up from the beginning to maintain consistency. This simulation protocol is widely used in previous model optimization studies (Peylin et al., 2016; Ma et al., 2024)."

**References:**

*Peylin, P., Bacour, C., MacBean, N., et al. (2016). A new stepwise carbon cycle data assimilation system using multiple data streams to constrain the simulated land surface carbon cycle. Geosci. Model Dev., 9(9), 3321-3346. doi:10.5194/gmd-9-3321-2016*

*Ma, R., Zhang, Y., Ciais, P., et al. (2024). Stepwise Calibration of Age-Dependent Biomass in the Integrated Biosphere Simulator (IBIS) Model. Journal of Advances in Modeling Earth Systems, 16(6), e2023MS004048. doi:https://doi.org/10.1029/2023MS004048.*

**Comment #6**

Line 205-206: "An equilibration of the soil biogeochemistry was not needed because of absence of any effect on vegetation processes. ": I found this statement too much absolute. In principle the soil must influence vegetation processes through nutrient supply and the regulation on soil water process (transport, soil water holding capacity) that are influenced by soil biogeochemistry. Pls rephrase.

**Response #6**

We agree with the reviewer that soil biogeochemistry can influence vegetation processes through the regulation of nutrient supply and soil water dynamics. Previously, the soil pool was calculated by an analytical solution (Vuichard et al., 2019).

We have therefore revised the sentence on **L210–211** in the main text as follows:

"The equilibration of the soil biogeochemistry was calculated by an analytical solution (Vuichard et al., 2019)."

**References:**

Vuichard, N., Messina, P., Luyssaert, S., et al. (2019). Accounting for carbon and nitrogen interactions in the global terrestrial ecosystem model ORCHIDEE (trunk version, rev 4999): multi-scale evaluation of gross primary production. Geoscientific Model Development, 12(11), 4751-4779. doi:10.5194/gmd-12-4751-2019

**Comment #7**

Fig. 2 has a wrong caption when indicating the subpanels. Correct.

**Response #7**

We apologize for the mistake. The caption has been corrected as follows:

"Figure 2: Spatial patterns of AGB, GPP, and mortality rates from the observations (a-c), from the model with spatially constant parameters (d-f), and from the model with spatially varying parameters (g-i)."